# Health workers and Sub Saharan African women's understanding of equal access to healthcare in Norway

**Inger-Lise Lien**⊙*

Norwegian Centre for Violence and Traumatic Stress Studies, Nydalen, Oslo, Norway

* i.l.lien@nkvts.no

## Abstract

This article describes and analyzes conceptions of equal access to healthcare by health workers and Sub Saharan African women living in Norway. The main objective of the study was to find out if there is equal access to healthcare as understood by both the provider and receiver side of healthcare. The two sides have different positions from where to observe and judge the services given, which can give a broader understanding of the healthcare system. Do Sub Saharan African women find healthcare services unjust and discriminating? Do health workers share conceptions of access held by these women? This study used a qualitative fieldwork research design. One hundred interviews were done with health workers and 55 interviews were done with Somali, Gambian and Eritrean women who all had experienced female genital mutilation/cutting (FGM/C). The study found a mismatch in the conceptions of access to healthcare between health workers and the women. Health workers did not believe there was equal access to healthcare and were critical of how the system functioned, whereas the women trusted the system and believed there was equal access. However, both sides had corresponding views on the following challenges facing the healthcare system: little time available to identify symptoms, difficulties in navigating the system, difficulties in getting referrals, and some negative adjudication by some health workers. Bourdieu's concepts of field, habitus and hysteresis, and candidacy theory were used to analyze the collected data. It was concluded that health workers and the women based their experiences of healthcare on differing cultural frames and expectations. The women seemed to base their assessments of healthcare on previous experiences from their home country, while health workers based their understanding from experiences within the system.

## Introduction

In Norway, equal access to high-quality services is mandated in the Patient and User Rights Act, and it is argued that this should promote trust [1]. A comparative study of trust in healthcare systems in several European countries, found that Norway in certain areas scored relatively low compared to other European countries [2]. Other studies have posited that a general

**Data Availability Statement:** The raw data is based on qualitative interviews that contain very sensitive information about informants, both physical and psychological health as well as health

seeking behaviour, and it is a possiblity for indirect identification of persons. Within the application for ethical approval, we have promised the ethics committee that we will secure the raw data we collected, as it is very sensitive. We have also told our informants that they will be given anonymity, and that the sensitive raw data shall be protected from sharing and accessing. The relevant data that is presented in the submitted manuscript, is within the article. Contact information for data access request are Norwegian Centre for Violence and Traumatic Stress Studies, tel: +4722 59 55 00, e mail: postmottak@nkvts.no.

**Funding:** The Norwegian Research Council has financed the main part of the research. NKVTS with support from the Ministry of Justice has contributed to the financing of the research. No - The funders had no role in study design, data collection and analysis, decision to publish, or preparation of the manuscript.

**Competing interests:** The author have declared that no competing interests exist.

lack of trust in a society can impact the use of healthcare [3,4], thereby leading to a negative spiral of both trust and the use of healthcare services. However, immigrants living in Norway, who seem to have worse living conditions than the majority of the population, still seem to have a high degree of trust in political, social and healthcare systems [5]. On the other hand, studies have also highlighted differences in the use of emergency services (ER) and the general practitioner (GP) [6]. Immigrants from Somalia use ER services more often than Ethiopian, Eritrean and Gambian immigrants [6–8]. Næss [9] argue that there is a lack of trust among the Somalis and argue that "cramped timeslots and a cut-to the case attitude among doctors" can be a poor foundation for trust", thereby placing reasons for distrust into the way that the services are organized.

This article analyzes how equal access to healthcare is perceived by Sub Saharan African women (SSA women) from Gambia, Somalia, and Eritrea and by health workers in Oslo. The article is part of a project called "Physical and psychological healthcare for girls and women with Female Genital Mutilation/Cutting (FGM/C)" that has been financed by Norsk Forskningsråd (the Norwegian Research Council) and Norwegian Center for Violence and Traumatic Stress studies with support from the Ministry of Justice.

Studies of access to healthcare has through years shifted its focus from the supply side of access to the demand side [10]. According to Andersen et al. [11, p.51], access to healthcare is defined as "those dimensions which describe the potential and actual entry of a given population group to the healthcare delivery system." Aday and Andersen [12] and Goodhard and Smith [10] have argued that equity of access exists when need rather than structural or individual factors determine who gains entry to the healthcare system. In a review of the literature on healthcare access, Goodhard and Smith [10] criticize the reviewed research on access for being weak in terms of theoretical frameworks. They distinguish between horizontal equity which requires equal treatment for individuals with equal needs, and vertical equity which requires individuals with different needs to receive different amounts and levels of services.

In recent years the focus has shifted more to the demand side. In an Italian study, Glorioso and Subramanian [13] analyzed the use of healthcare services by rich and poor patient groups. They argued that the benefits of prevention and early diagnosis are unequally distributed in favor of the wealthy. When health status is perceived as good, poorer groups tend to limit their use of expensive or inefficient health services; that is, they limit their use of preventive care and postpone their access to curative care, creating conditions that may harm their future health status.

Ruger [14] who focuses on the barriers that may reduce equal access to healthcare, chose to use a health capability perspective that focuses on users' resources. This way of thinking has been supported by others who are concerned with empowerment strategies and also the degree of fit between the demand and supply sides.

Macintyre et al. [15] have identified the need for a systemic analysis that can inform policymakers about the "fit" between needs for healthcare and the receipt of healthcare, taking both sides into account.

"Access, or empowerment to use healthcare services will only be achievable if all dimensions of access are addressed and both the healthcare system and individual perspectives are taken into account" [15 p. 190]. This focus on "fit" and "gaps" between the provider and the receiver has increased interest in more closely examining the receiver side of healthcare, not only individually, but also from cultural and symbolic perspectives. New analytical concepts to capture resources on the demand and receiver sides have been developed, such as empowerment, health literacy [16], health capital [17] and candidacy [18]. In this article, candidacy theory [18] will be applied as it tries to bridge the two sides by introducing the formulation of "*joint negotiation*". In addition, Bourdieu's concepts of field, habitus and hysteresis [19–21]

will be applied complementary to candidacy theory as it widens up and strengthen the analysis of access to healthcare.

The aims of this article are to present attitudes held by both the providers and receivers of healthcare and see how health workers and SSA women view equal access of healthcare for women who come from countries practicing FGM/C. Research questions include the following: Is there consensus or disagreement between the two sides? Do SSA women find the healthcare services to be unjust and discriminating, or do they have trust in the system? Do SSA women seek alternatives in other countries or alternative treatment in their original home countries when they have healthcare needs, or are they satisfied with the healthcare that is provided by the healthcare system?

The article is organized to provide the following; 1) an introduction to candidacy theory of healthcare negotiation [18] as a mesa-level theory, and 2) an overall theoretical framework derived from using Bourdieu's theory of field, habitus and hysteresis [19–21]. Moreover, this article 3) introduces the empirical findings arranged according to the two theories and 4) analyses and discuss the split between health workers and the SSA women's perspectives on equal access based on Bourdieu's concepts of hysteresis field and habitus.

## Materials and method

### Analytical frameworks

Dixon Woods et al. [18] have defined candidacy as the following: "... the ways in which people's eligibility for medical attention and intervention is *jointly negotiated* between individuals and health services" [18 p.7]. Candidacy has seven dimensions that, according to Dixon-Woods et al. capture the characteristics at play during this joint negotiation [18,22]. These dimensions are 1) *identification*, which refers to users recognizing symptoms and health needs; 2) *navigation* by healthcare users, which requires an awareness of the services available; 3) *permeability* of services, which encompasses accessibility of the services (may include competence to apply); 4) *appearances* at the health services (implying credibility, or ways of formulating the healthcare need in a credible way); 5) *adjudications*, which are judgments made by professionals from their interpretation of their clients' needs; 6) *offers and resistance* to reference by the patients as well as their reactions to offers of medication; and 7) *operating conditions*, which are locally specific influences on interactions between practitioners and patients.

The seven dimensions strongly emphasize the users' ability and work to formulate and present their claim for healthcare. On this receiver side of healthcare, there can also be resistance to the treatment provided. People in need of healthcare may struggle with developing a claim, which is related to the identification of symptoms, knowledge about available services, and lack of competence in health issues or lack of economic ability to purchase the medication prescribed. Healthcare providers must make adjudications that influence the end result of the healthcare negotiations and claims. The concept of negotiation captures many aspects of acting and communicating that occur before the healthcare-seeking process is initiated and claims for healthcare are made, as well as aspects that occur after (like resistance to using medicine). The concept of negotiation used in the theory is therefore a very broad concept that has encapsulated considerations and experiences developed by the patient over a life long process.

In reviewing and analyzing the candidacy theory, van der Boor and White [23] discovered that in their review, there were more data on barriers to access than on service negotiation. They encouraged the use of candidacy theory since it considers many relevant factors but captures asymmetry in power and an enforcement of dominant values onto service users. They also found that their patient group had experienced many barriers to accessing and negotiating medical health services, and the greatest problem during negotiation was language. Koehn

[22] used the candidacy framework in a study of ethnic minority seniors' access to healthcare in Canada. Koehn argues that the first six dimensions in the theory deal with negotiation, but the seventh dimension represents the environmental context in which the negotiation occurs; thus, this dimension lies beyond negotiation [22].

To present a theory as a contextual frame around the dimensions within candidacy theory that lies outside of "negotiation", Bourdieu's theory of practice seems as an appropriate complimentary contribution [19–21]. Bourdieu' theory operates on a meta-level and can capture differences between groups of people at a wider and more abstract scale than that of the individual, through the use of the concepts of field, habitus and hysteresis.

According to Bourdieu, a field is a place or an overall structure where competition and inequality of power affect one's struggle for position and access. Attached to the field is the habitus, an internalized structure that consists of codes, values, tastes and distastes, schemata and memory traces that result from experience, habits and socialization that occur in the field [20,21,24]. The habitus is embodied and can also be unconscious in the way that it is taken for granted. Through their actions and communication, individuals may reveal their taste and values and emit signals. People in other fields and with other habitus, can use these signals to make distinctions and either exclude these individuals, putting them in disadvantaged positions [20], or include them.

Hysteresis is a concept that Bourdieu [19] applies to identify mismatch between field and habitus that occurs in situations of change and transformation. Hysteresis implies that something is "lagging behind". The field can change, increasing the misalignment between the habitus and the new, changed field. When immigrants move to another country, their internalized habitus can be in mismatch with the new situation or structure in their new location. This situation is a hysteresis condition because their habitus lags behind and becomes obsolete, which can lead to unfavorable decisions:

> "The presence of the past in this kind of false anticipation of the future performed by the habitus is, paradoxically, most clearly seen when the sense of the probable future is believed and when dispositions ill-adjusted to the objective changes because of a hysteresis effect are negatively sanctioned because the environment they encounter is too different from the one to which they are objectively adjusted

> [19, p 62].

The habitus concept has been criticized for being deterministic, and not sufficiently capable of describing change. [24,25 p.159]. However, abstractions are necessary as tools for thought [26], and change is something that Bourdieu captures using the concept of hysteresis. A defense for using Bourdieu's concepts of habitus is that it can analytically frame theories, like the candidacy theory, and thereby incorporate an understanding of differences and nuances of individual and group processes and negotiations, at a lower level. Due to migration, groups of people with different values and opinions are moving into different corners of the world, but all the same, grouping together into networks in the diaspora, having many similar habits and identities in spite of the differences between them. Migrants may create their own fields in the suburbs of cities where traits from the old habitus continues, or they may develop a new habitus. The habitus concept can capture the meaning making dimensions for groups of people that exist within these new field structures.

The medical healthcare system can be seen as a separate field. Experiences from an old medical habitus can exist in peoples mind and be in mismatch with a new developed medical field, as has been described in a study from Denmark [27] using Bourdieu's analytical concepts

of hysteresis and habitus. This study found that the healthcare field has changed over years into an efficient and economically oriented market model. Danish families of stroke patients are currently expected to provide more caring tasks than was expected in earlier times, like in the 1960s, when the healthcare and welfare systems were developed. The relatives of Danish stroke patients, who were accustomed to an earlier medical habitus, therefore expected the current healthcare system to perform most of the caring tasks rather than leaving them to relatives or outsiders. This unfulfilled expectation led to irritation and disappointments. There was hysteresis between relatives' internalized habitus from earlier and the current medical field and habitus (i.e. the way the medical has retracted from many caring tasks it used to perform, leaving them to relatives).

Bourdieu's concept of hysteresis demonstrates that there is more to interaction and communication between the healthcare system and patient groups than what the candidacy theory alone endorses. Behavior based on an older habitus that does not match with standards for candidacy held by health professionals today, who exercise their adjudication within a modern Western healthcare system, should not necessarily be understood as a failed individual strategic negotiation process. Rather, it may be a condition of hysteresis with standards and judgments of candidacy belonging to a medical field with its habitus that new patient groups have little knowledge of since their own habitus can lag behind or be attached to a different medical field.

In the analysis of the empirical findings of this study, dimensions within the candidacy theory are introduced and framed using the concept of field, habitus and hysteresis to determine whether there is a match, mismatch or hysteresis between understanding of access to healthcare in health workers and women from immigrant communities, especially those from the SSA countries Somalia, Gambia, and Eritrea.

## Study design and procedure

This study is based on qualitative data collection in Oslo involving participants on the supply and demand sides. One hundred interviews were conducted among health workers who were asked about equal access to healthcare for SSA women. The participants were general practitioners (24), psychologists and psychiatrists (9), gynecologists (4), neurologists (4), midwives (19), nurses (8), sexologists (3), and administrative health personnel (3). Participants were asked if, according to their experience, SSA women received healthcare access, equal to that of the general population, and were asked to described and analyze their experiences in general, and as they related to the consequences of FGM/C. The interviews each lasted for 1–2 hours, and some participants were interviewed twice. While nurses and midwives were easy to recruit for interviews from both hospitals and health centers in immigrant-dense areas, it was difficult to arrange meetings with the GPs who have a heavy work load of, on average, 56 hours per week [28]. Emails were sent to several medical centers in Oslo, and telephone calls were made, but without any result. The solution was to recruit GPs using colleagues' networks. A research assistant had an uncle who was a GP in Oslo who helped with recruitment, but other entry points to the professional community were necessary. The lead researcher contacted GPs in immigrant-dense areas who came to our assistance. This snowball strategy [29] and targeted method [30] were also used to recruit psychologists and psychiatrists. These methods are options when studying populations that are difficult to reach, like drug addicts and others, but may also be used to reach professional communities that are somewhat unavailable. Leveraging connections made through acquaintances and key participants is a qualitative method commonly used by social anthropologists, and it implied borrowing the trust that other GPs

had in the professional community. The interviews were conducted in clinics, coffee shops close to the clinics, or private homes.

In addition, 55 SSA women—20 Gambian, 20 Somalian, and 15 Eritrean women—were interviewed about their experiences with the Norwegian healthcare system.

The SSA women came from countries in which FGM/C occurs, although specific practices and thus the consequent complications varies.

The World Health Organization (WHO) defines FGM/C as "all procedures that involve partial or total removal of the external female genitalia, or other injury to the female genital organs for non-medical reasons." [31].

An Eritrean midwife helped recruit Eritrean women, mostly through her own Christian Tigrinya network. One Gambian woman, an activist and a member of the project's user group panel, participated in the project as a connection to the Gambian Mandinka community and conducted some interviews with the Gambian women herself. A Somali midwife in her 50s was willing to assist in recruiting and interviewing Somali participants, making use of her substantial network in the Oslo area. The anthropologist conducted interviews either with an assistant or alone with women within all three ethnic groups. All interviewees received both oral and written information about the project and signed a letter of consent. The interviews lasted approximately two hours, and several of the participants were interviewed twice. Interviews were not audio recorded since the participants were not comfortable with audio recording. Handwritten notes were taken during the interviews and later digitally transcribed and analyzed. The Ethical Review Board (REK) in Norway approved this study on September 12, 2017.

## Results

### Health workers' habitus and perspectives of equal access to healthcare for SSA women

The GPs, nurses and midwives were mostly women aged between 45 and 50 years, with an average age of 48 years. Most of participants had more than 10 years of work experience, but some had more than 20 years. Out of 24 GPs there were only 7 males, and the oldest was 69 years old.

All health workers were asked if SSA women receive equal access to healthcare. The immediate response was negative. However, health workers continued their reflections by saying that in principle all immigrants should have equal access to healthcare, but in reality they do not receive it. They argued that the medical system is hierarchical and bureaucratic and meets clients on its own terms rather than on those of the patient. The system is to blame for being rigid and not giving patients enough time, competence, and understanding. Some health workers said that the system discriminates between those who have resources and those who do not. Others said that doctors and nurses are too impatient, jumps to conclusions, and do not provide sufficient consideration for patients who do not speak the official language and have little health competence. Health workers talk behind patients' backs and stigmatize them; therefore, women with immigrant backgrounds such as the SSA women do not receive equal access to healthcare. Below are some of the responses:

> "No, they get three-quarter services. The immigrant women must wait longer because they are more difficult to treat. They change their GP a lot, and I think it is because they do not get the treatment they want. Some do not show up for their appointment or do not pay, and this means they are not wanted as patients."

> (GP, female, 63)

"In Norway you have to stand up for your rights, and they do not know their rights."

(GP, female, 52)

"No, there is lack of cultural understanding in the health services."

(Midwife, 61)

"The doctors don't give an offer that fits with the background of the patient."

(GP, female, 45)

"Sometimes seven people arrive for consultancy when there should only be one patient. There are lots of prejudice against them, but some of the prejudice is real. You have to be realistic."

(GP, female 50)

"They express their illness in a diffuse way, and it is difficult to understand what they say."

(GP, female, 43)

"There is no equal access. Our system is complicated, and it is difficult to navigate. The use of interpreters makes the situation difficult also. When we have consultancy with an interpreter in the room, it takes double the amount of time."

(Midwife, 58)

"No, they are discriminated against at all levels of the society."

(Sexologist, 48)

Arguments from health workers explaining why there is not equal access to healthcare in the Norwegian health system are included in the quotes above and in Table 1. Some health workers blamed the overall system's structural, hierarchical and bureaucratic function. Second, health workers blamed unequal access on attitudes held by health workers (i.e., on their own habitus). Third, health workers blamed SSA women for their manner of behaving and thinking, their lack of knowledge about the health system in Norway, their lack of competence and education about the body and health, and their lack of knowledge about their rights. Health workers' conceptualization (habitus) of SSA women's communication of needs can be categorized and summarized as the health workers' "ideas about SSA women's habitus".

The morality within the medical habitus of health workers seems to be based on a human rights conception, and health workers often complained that SSA women did not know their human rights and therefore did not claim their rights. Health workers could identify the mismatch between their own thinking and that of the SSA women, who may reason and behave in ways different than what they expected, ways of behaving and thinking that, in this work, are categorized as lying within the SSA women's habitus.

Health workers think that SSA women struggle with navigation, permeability, and appearances that can be linked to the fact that they are poor refugees and migrants who have other ways of diagnosing their illnesses (i.e., according to Bourdieu's terminology a different medical habitus). As candidates for healthcare, they are seen as lacking competence and system understanding, and due to language problems they are seen as unable to properly communicate their health needs. The candidacy dimensions developed by Dixon-Woods et al. [18] are

**Table 1. Health workers' perspective and habitus on reasons for unequal access to healthcare.**

| 1.Blaming the overall medical system—the field | | 3. Health workers blame "SSA women's habitus". | Candidacy dimensions |
|---|---|---|---|
| Hierarchical and complex system where those with resources receive more Bureaucratic system System does not meet the individual at his or her premises Systematic lack of time to determine diagnoses. | | • Come for unnecessary problems, afraid of high fever • Have poor body and health knowledge • Provide strange descriptions of symptoms | *Identification* |
| **2. Health workers blame health workers'own habitus** | **Candidacy dimensions** | | |
| • Underestimation of immigrants • Conscious and unconscious stereotypes • Stigma • Quick rejection of patients' problems • Health workers overly strong or weak cultural sensitivity. | *Adjudications* | • Have weak system competence • Use the system incorrectly • Do not arrive on time • Do not know their rights • Have insufficient networks to help them | *Navigation* |
| • Higher barriers for immigrants to be referred • Resistance to making extra effort for immigrants | *Permeability* | • Disturb consultancy by having an interpreter in the room • Distrust the doctors • Distrust the system • Are afraid of being reported to child welfare system | *Permeability* |
| • Ineffective use of interpreters • Use of family interpreters restricts information sharing • Slow and difficult communication | *Appearances* | • Have language problems • Will sometimes refuse to take clothes off • Cause irritation • Do not make demands | *Appearances* |

formal descriptions, that can be filled with substantive contents. The substantive contents have been added to the left side of the Table 1, and the formal descriptions within the candidacy theory to the right side. Adjudications include both stereotypes, underestimation, stigma and rejection as interpreted by the health workers themselves. Permeability issues involve resistance towards referrals of these women further in the system. When health workers were asked about the substantive content of the navigation dimension of the SSA women's habitus, they mentioned weak system competence, incorrect use of the system, weak networks and not knowing their rights so that they can make adequate healthcare claims. These findings indicate that health workers can assess the hysteresis that Bourdieu has described, that is, a mismatch between the habitus and the requirements of the field. The overall conclusion of the health workers is that there is not equal access to healthcare because of these systemic factors, the mismatch between health workers' and the SSA women's ways of acting and thinking, their habitus.

## SSA women's conceptions of equal access to healthcare

The average age of the Gambian women was 41, the Eritreans 46, and the Somalians 43.

The youngest participant was a 22-year-old Somali woman, and the oldest, a 68-year old Somali woman. In relation to age and gender, there was not a big difference between health workers and SSA women, as most of them were women in their 40s and 50s. The SSA women had migrated to Norway either as refugees or as wives of migrants. None of them were born in Norway.

When asked if they receive equal access to healthcare, the SSA women immediately answered affirmatively and then added that they are very satisfied:

"I have 100% confidence in the Norwegian healthcare system"

(Eritrean, 55).

"The Norwegian healthcare system is the best; I only have good experience"

(Somali, 42).

"They don't say that you are African and we have to take Norwegians first. No, no, no. It is equal. We get the same treatment and priority as the Norwegians. I am satisfied"

(Eritrean, 53).

"I have had two GPs. The first one I could have married, and the second one could have been my sister"

(Gambian, 48).

"The Norwegian health system is first class"

(Gambian, 36)

"The Norwegian system has good hygiene, good medicines, and nice doctors"

(Gambian, 23).

As an example of how excellent the Norwegian healthcare system is and how generous the GPs are, a Gambian woman explained that when she divorced her husband, her GP even came to her home, talked with her, helped her to apply for social security income, and ensured that she had sick leave for many weeks. She said that she loved the GP.

However, 2 of the 55 interviewees, had heard of discrimination against immigrants, though they had not experienced it themselves. One woman said that she thought there was discrimination in the healthcare system and that she had experienced it herself. These comments were exceptions. The interviewees generally discussed the healthcare system positively and with a great deal of respect and trust. Principally and as a moral imperative, they expressed a deep trust and confidence in the health system in Norway and especially in their GPs.

However, there is another level in the SSA women's analysis of the healthcare system. After first expressing their belief that they received equal access to healthcare and their confidence in and respect for the Norwegian healthcare system and its health workers, the SSA women complained about several issues:

"The waiting time for consultancies is too long"

(Eritrean, 53).

"The GP does not ask us about FGM/C and does not understand the problems that this procedure causes"

(Eritrean, 53).

"If the doctors examine us in our private areas, they do not discover that we are cut or mention that we are cut"

(Gambian, 50).

"They do not understand those of us who are circumcised"

(Gambian, 35).

"Some of the GPs do not have a lot of competence because they use the Google a lot. I could treat myself by investigating on Google. That is what the doctors do"

(Somali, 48).

"In Africa, it is the doctors who decide. In Africa, doctors take responsibility. In Norway, when you are ill, doctors want to collaborate with you. It is as if they do not care about you. They ask you, 'What medicine do you want?'"

(Somali, 59).

Eritrean, Gambian, and Somali women generally found the health system to be fair and sufficient. This goes for all the three ethnic groups. However, they stated that doctors were sometimes insecure about their own competence, consulted with books or Internet, and asked the patients questions, so the patients thought the doctors did not have sufficient knowledge. The women's understanding and assessment of the healthcare system also seem to be based on a comparison of the system in Norway to that of their original homeland rather than against ideologically based principles about rights grounded in policy documents and moral principles, and they all concluded that the Norwegian system is better. In this way their medical habitus based on their home country, was used as a baseline to assess the medical system in Norway. The SSA women do not prefer the medical system in their home country. They do not call relatives and friends in other countries to give them advice when they are ill, nor do they express a desire to seek help from medical systems in other European countries. However, on another level they mentioned deficiencies and weaknesses in the Norwegian medical system. Below is a table presenting a list of positive attitudes and complaints that were expressed during interviews.

In this section of research, the candidacy theory is used somewhat different than in the Table 1. The dimensions described are usually thought to be based on the user group's claims during negotiation. In Table 2 it is reversed, it is the SSA women's thoughts and ideas about health workers identification, appearances, permeability and adjudication of services that are assessed.

One of the most prevalent complaints is that FGM/C is not well understood and is not often mentioned:

"The experience with the healthcare system has been good but lacking in regard to conversations around FGM/C. It seems to be as taboo here as in the Gambia. It was never mentioned"

(Gambia, 48).

"I felt very much discriminated against by the doctor. I had no good communication with him. They just write and write and do not have the time, just 10 minutes, and then they take a lot of money from you"

(Somali, 59).

This is an organizational and structural analysis conducted by the women themselves that focuses on the time doctors have and ways of making diagnoses and referrals. Here, health

**Table 2. SSA women's conception of the system.**

| 1. SSA women's overall conception of the system—the field | |
|---|---|
| The system is hygienic, competent, has skilled doctors, proper medicine, and does not discriminate between ethnic Norwegians and others | |
| Health workers are usually very nice and helpful | |
| **Blaming the system** | |
| There is not enough time within the system | |
| The waiting time to receive specialist services is too long | |
| Consultation is costly | |
| Communication is difficult | |

| 2. SSA women blame health workers' medical habitus | |
|---|---|
| • GPs sometimes lack competence<br>• GPs can lack understanding of SSA women's problems due to FGM/C<br>• GPs do not always determine what is wrong with the patient<br>• Some GPs jump to (the wrong) conclusion too quickly | *Thoughts about identification of symptoms by health workers* |
| • Some GPs rely on Google a great deal<br>• They often do not ask women about FGM/C<br>• They do not find the reasons for pain<br>• They do not excel at reading journals<br>• They do not always have adequate routines | *Thoughts about appearances of health workers* |
| • Some do not refer patients for further treatment | *Thoughts about appearances of health workers* |
| • They think patients have pain because they do not want to work<br>• Some GPs underestimate patients<br>• Some GPs distrust patients' intentions | *Thoughts about adjudications made by health workers* |

workers perceptions align with those of the SSA women. However, patient involvement which is usually seen as the democratic right of patients to participate in decision making, is not necessarily viewed as something positive and in line with the women's habitus and expectations of the doctor. Instead, it can be viewed as a weakness of the system, which is a condition of hysteresis between different habitus.

## Unexplained pain

Another issue that deals with health workers' identification of symptoms, is that many women have pain that has not been diagnosed. Eritrean, Somali, and Gambian women mentioned this often:

"Why is it that you have pain, day in and day out, and the GP tells you that there is nothing wrong with you?"

(Somali, 50).

"My pains kill my brain."

(Gambian, 54)

"I do not understand that when I go to a GP and tell him that I have pain, he explains that I am not ill. Then I think I must experience pain the rest of my life. I must just get used to

pain. But I also think that this pain is not normal, that the GP is stupid, that there is a lot the doctors do not know."

(Somali, 59)

"I have confidence in the health system and do not think they discriminate. But they must do more examinations and take people more serious when they have pain. They cannot just say 'Take it easy; everything is fine.'"

(Gambian, 48)

Out of 53 women who discussed their pain during the interviews, 36 women said that they suffer from pain and illnesses. The pains are often around the reproductive area as well as in the legs, back, and stomach. Eleven women suffered from pain that had been attributed to a diagnosis. Twenty-five women said they suffered from pain that was undiagnosed and could not be explained by the GP.

Several of the Gambian women thought that their chronic pain was due to an FGM/C operation performed many years ago. Seven Gambian women out of 20 (35%) argued that they had problems with their reproductive system as well as pains and dryness during sex, and according to some of them, this could be caused by female genital mutilation. They linked their pain as well as their depression to the effects of having been cut many years ago.

"I go to the doctor all the time, and one doctor called it referral pain due to FGM"

(Gambian, 54).

"When I sit at a bus stop and it is cold, I will get urinary infection. It is because my labia has been removed during circumcision and do not protect me from cold. I get antibiotics. I experience pain when urinating, and then it is difficult to have sex and you feel ill all the time. Then you become depressed"

(Gambian, 40).

When the medical healthcare system does not provide help for psychological and physical suffering and pain, women can turn to spiritual healing rather than seek help from medical systems in other countries. Many of the women explained that they did not believe in spiritual healing and had not used imams, priests, or marabous to help them when they were ill, but others had contacted religious leaders in their previous home country to help them. These women ordered amulets, Qur'anic verses, beans, and holy water from abroad, and made telephone calls to spiritual healers. In some instances, holy men (called "marabou" in Gambia) performed ritual slaughtering of a goat or chicken, usually in Africa after orders were sent via telephone or SMS. It seems that spiritual healing is particularly effective for alleviating the symptoms of mental health issues. Some interviewees reported feeling much stronger and had an increased hope of recovery because the spiritual healers would pray to God and try to improve their health or destiny via different rituals performed in the original country and in Norway. Spiritual healers can be found on certain streets in Oslo. Explanations for suffering could be the evil eye, witchcraft, or jealousy that other people feel toward them, and these afflictions push individuals to seek protection and to remove black magic that could have been put on them or djinns that are troubling them, causing them to feel pain and receive ill fortune.

Each of the three groups has its own traditions that include an inbuilt healthcare system that is connected to the religious practice. Eritreans are connected to the Pentecostal church and believe in praying and laying on of hands. Every Friday in the Pentecostal church there are prayers for those who are ill and have pains.

The Somalis also seek healthcare within their religion, and they distinguished between going to an imam, a religious healer, or an Islamic conference outside the country. Some imams or healers are believed to have more ability to cure than others.

"I have a lots of pain. Therefore, I go to the Mosque and pray."

(Somali, 48)

"If a Somali woman has a psychological problem or pain, she will not go to a psychologist, because that means she is nuts. She would rather go to an imam that tells her that there is a djinn within her. He will then try to remove the djinn."

(Somali, 41)

"To go to the mosque is first choice. The second choice is to go to Sweden in the summer where there is a big Islamic conference. People go there to get healed. Swedish imams, and especially English imams, are seen as more competent than the Norwegian Imams. The imams do not demand payment, but people must pay according to their ability and desire."

(Somali, 35)

The SSA women thereby have alternatives based within their traditional habitus they use when navigating their personal healthcare. This practice is an extension of their medical field and habitus that includes more options for healthcare and health treatment than the medical system can provide. Some of the SSA women seem to resort to religious healing when they feel that the medical system does not provide a sufficient treatment or further referrals when they have pain.

## Discussion

Health workers have an *internal position* in the medical system from which they can observe the interaction that occur between health workers at different levels and between the patients and the system. They hear arguments and comments that are passed during both formal and informal circumstances surrounding receiving, treating, or excluding patients. In addition, their job positions are based on medical and health studies that involves courses on healthcare, human rights, and medical ethics. With this background, combined with experiences and observations, health worker participants in this study concluded that there is not equal access to healthcare for women from SSA backgrounds. These health workers presented arguments that assigned blame for this inequity to the system, how it is organized, the attitudes of health workers, as well as the SSA women's own behavior and forms of communication, that is, on their habitus. Health workers have observed some mismatch in expectations of SSA women and Norwegian medical practices. The health workers are disappointed with the system since they may have an ideologically based high expectation for justice within healthcare, and they therefore do not believe there is equal access to healthcare according to their own observations and assessments.

The SSA women have an *external position* from which they seek access to the system. At the principal level, the SSA women seem to trust the overall system's overall ability to provide

diagnoses, referrals, treatment, and healthcare. Their expectations do not seem to be based on the formulation of rights as an abstract conception from where they measure their achievements to healthcare, making them disappointed with what they receive. In fact, they seem to think that the system and the healthcare workers provide them with a kind of help they would not have received in their original home country. There seems to be a mismatch and hysteresis between the SSA women and health workers perception of the situation. The SSA women have had both a positive healthcare experience and general fulfilment of their healthcare expectations, since they are accustomed to medical care in the African continent that may not equal the care they receive in Norway. There is a clear gap, mismatch and hysteresis between the demand and supply sides of the medical system in terms of the two groups' overall understanding of equal access to healthcare for SSA women. With their outsider perspective, the SSA women can be ignorant about discriminatory factors on the supply side, because exclusion factors are invisible to them. As such, very few women reported that they have experienced discrimination from health workers when they have been in need.

However, on *a practical level*, there is a match between the two parties' understanding of the perceived weaknesses of the system, such as limited available time, a shortage of opportunities to explain healthcare needs at the GPs' offices, insufficient use of interpreter, communication problems during consultancies due to cultural and language differences, and a lack of referrals to specialists. Some of the SSA women feel that there can be barriers to further referrals, and this shortcoming is also perceived by health workers. The parties agree that there are some unfortunate deficiencies within the system, even though the two parties also stress different aspects, such as issues of competence: health workers place blame on the SSA women for their lack of health competence while several of the SSA women think that some doctors must be incompetent when they cannot find a cause for pains and do not have the competence and willingness to discuss FGM/C. When the SSA women fail to receive a diagnosis, are disappointed with a diagnosis, or are not cured, they sometimes turn to spiritual healing, which can be describe as returning to their traditional habitus.

According to Kleinman [32], physicians have a narrow focus on diagnosis and treatment (disease), which leads them to exclude the patient's experience (illness) from their consideration. When medical doctors do not manage to provide a diagnosis or treatment, patients may search for alternatives within the spiritual realm, implying that their management of suffering and their illness widens their medical field and habitus, and spiritual healing becomes more prominent focus. Alternatives like these are often seen as illegitimate by medical experts and represent a hysteresis condition to the medical field. However, patients from an African background may use these alternatives in combination with the medical system to manage their health. These alternatives can be viewed as complementary to the medical system, even though they function according to nonscientific criteria and thinking and are often part of an older habitus or a habitus from a more spiritual realm.

## Conclusion

The hysteresis effect and mismatch apparent in the conceptions of equal access to healthcare between health workers and SSA women, has mostly been analyzed in terms of Bourdieu's concepts of habitus and field. In addition, Dixon Woods' theory of candidacy has been presented; this theory uses "joint negotiation" to explain the driving forces of access to healthcare.

However, the concept of negotiation, with the attached dimensions, used within candidacy theory, is broad and goes further than the negotiations that occur in a consultancy room, as described by Sandman [33] and Sandman and Munthe [34]. In a consultancy room people bring relevant issues that they formulate, and at the same time they give out health signals that

health workers interpret. In the candidacy theory the negotiation process is much broader including both actions, knowledge, mindset and experience that may have its base long before the meeting with health professionals. In this way, the concept of negotiation transgresses into the reality of the habitus of the patient groups, on the supply side. By doing so, it touches upon very relevant factors within their habitus, but without being framed analytically into a coherent structure of internalized codes, understandings, and values obtained both consciously and unconsciously within a long socialization process. These frames, in this study called "habitus" can be diverse due to the different people and fields, and they can match or be in mismatch with different fields, structures or people involved within healthcare on the supply side. Candidacy theory as an analytical instrument can explore the habitus of different groups, but cannot successfully be used to frame a coherent habitus.

Through the use of the negotiation conception with all seven elements included, the importance of the demand side seems to outweigh the importance of the supply side in order to get access. According to the theory, negotiation should be *jointly*, but in reality, it is the adjudication done by the health workers that in the end provides the ultimate power to decide candidates' entry. This is what interviews with health workers in our study stressed. These health workers have been aware of their power as well as the SSA women's lack of power and knowledge about rights. Within the candidacy theory, power relations are blurred and the blame for unequal access to healthcare is to a great extent assigned to the demand side, even though responsibility is theoretically thought to lie in the middle as a "*joint negotiation*."

Power relations within healthcare are not necessarily visible to user groups who are new to a country and have little competence and scientific knowledge about the healthcare system. If their communication about healthcare needs is based on an old habitus, with differing ways of expressing symptoms and needs, this can lead to hysteresis between existing criteria for access and people's behavior and bodily signals. Medical health personnel can, on the basis of the signals they receive from the habitus that other people carry, permit or deny access; thereby, they may also be able to observe the hysteresis that exists between a patient's habitus and the habitus of the medical field. This observation is based from within and is a benefit for assessing equal access to healthcare that the patients trying to obtain access do not necessarily have.

In recent years the medical sector has shifted from paternalism towards a more democratic healthcare approach. Most patients are expected to have a high level of education and extensive knowledge about health issues—so much so that they are welcome to have a say when it comes to medication and healthcare.

Complaints from the SSA women directed towards the healthcare system involve structural and organizational matters. In addition, SSA women find Norwegian health workers have limited competence when it comes to FGM/C. Many of these women suffer from unexplained pain that needs to be cured. SSA women should not only be offered reproductive and sexual counselling, but also receive pain management and treatment.

New patient groups may have less schooling and scientific knowledge about health issues and less knowledge about the system and their rights. Because of how it is organized and functions, the medical field is, consequently in hysteresis with the expectations of these new patient groups. As this study demonstrates, this misalignment can negatively affect the new patient groups' equal access to healthcare. Moreover, this unequal access to healthcare may not be apparent to new user groups, who may have a high trust in the system.

## Supporting information

**S1 Appendix. This is interview guide for health workers.**
(DOCX)

**S2 Appendix. This is interview guide for SSA women.**
(DOCX)

## Acknowledgments

I would like to thank Cecilie Knagenhjelm Hertzberg, Neneh Bojang, Safia Abdi Hasse, Sara Khasay and Sara Zorica Mitic for their contribution to the data collection process. Thank you to the Health Forum for Women in Oslo for access to some participants. I would also like to thank Mai Mahgoub Ziyada, R. Elise Johansen and Marianne Opaas at NKVTS for their support and contribution to the project, as well as professor Rune Halvorsen for his reflections, contribution to new literature as well as comments on the theoretical analysis, particularly Bourdieu's concept of habitus.

## Author Contributions

**Conceptualization:** Inger-Lise Lien.

**Formal analysis:** Inger-Lise Lien.

**Funding acquisition:** Inger-Lise Lien.

**Investigation:** Inger-Lise Lien.

**Methodology:** Inger-Lise Lien.

**Project administration:** Inger-Lise Lien.

**Validation:** Inger-Lise Lien.

**Writing – original draft:** Inger-Lise Lien.

**Writing – review & editing:** Inger-Lise Lien.

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
