## [Decision Letter · Decision Letter 0]

17 Jun 2021

PONE-D-20-39751

Unequal access to healthcare: a hysteresis condition?

Health workers’ and Sub-Saharan African women’s understanding of equal access to healthcare in Norway

PLOS ONE

Dear Dr. Lien,

Thank you for submitting your manuscript to PLOS ONE. After careful consideration, we feel that it has merit but does not fully meet PLOS ONE’s publication criteria as it currently stands. Therefore, we invite you to submit a revised version of the manuscript that addresses the points raised during the review process.

ACADEMIC EDITOR:

While the paper focused on an important issue, the paper required major re-organization for presentation. Specifically, the paper must start by following the journal instructions for each section. Please make sure to make a clear distinction of the results and tables addressing the aims as they must be included in the paper and not as supplement materials. Tables should have informative titles, what is presented, for who, where and when. Please see e-file with comments as well as  Reviewer 1's specific comments and feedback.

In addition, please follow the journal's requirements for qualitative study presentation.

We look forward to receiving your revised manuscript.

Kind regards,

Luisa N. Borrell, DDS, PhD

Academic Editor

PLOS ONE

Journal Requirements:

4. We note you have included a supporting information table to which you do not refer in the text of your manuscript. Please ensure that you refer to Table S3 in your text; if accepted, production will need this reference to link the reader to the Table.

Reviewers' comments:

Reviewer's Responses to Questions

**Comments to the Author**

1. Is the manuscript technically sound, and do the data support the conclusions?

Reviewer #1: Yes

2. Has the statistical analysis been performed appropriately and rigorously? 

Reviewer #1: N/A

3. Have the authors made all data underlying the findings in their manuscript fully available?

Reviewer #1: No

4. Is the manuscript presented in an intelligible fashion and written in standard English?

Reviewer #1: No

5. Review Comments to the Author

Reviewer #1: This article covers a very interesting and important topic. The study has many strengths since it collected data from a large number of health workers and women from SSA living in Norway, and extract useful conclusions. However, there are several issues related to the reporting and editing of the article that should be addressed.

Abstract:

- I suggest the abstract follows a common structure (background, methods, results and discussion), instead of including many subsections with specific titles.

Background:

- Please, read journal’s instructions on the information that background section should include. The structure of the background is not clear. I suggest starting with the 4th paragraph and ending the section by explaining the project in which the articles is included, and the objectives.

- The introduction presents results, please remove them from this section.

- It is not necessary to present and reference other articles published from the same project in the background.

Methods:

- Following journal’s instructions, the manuscript should contain sections on background, methods, results and discussion. The sections called “Equal access to healthcare” and “Habitus and hysteresis…”, thus, should be placed either in the background or in the methods section.

- My suggestion: create a general section called “materials and methods” where you can include these two sections on theory, then a subsection on “study design and procedures” (now called “Method”).

- I do not think it is necessary to report the age of the social anthropologists.

- Inclusion criteria and procedures to approach health workers for interview are not explained. Where were they recruited? How?

- Paragraph starting in line 266 reports information that should be placed in the results section.

- I suggest to delete this sentence: “Since she had done fieldwork among Somalis and Gambians in 274 earlier projects [32,33,34,35]”. It is unnecessary to justify who performed the interviews, and to reference all their publications.

- Information on purposive sampling should be better placed at the beginning of the methods section for the sake of clarity (now in line 281).

- It would be interesting to briefly describe the interview guides to help the reader understand which topics were covered in the interviews.

Findings:

- Tables 1 and 2 are cited in the text but they are supplementary files. As they are important to understand the results, I suggest including them in the main body of the article.

- Line 338: a table is cited but it is not clear which one.

- Please, use the same style always when quoting.

- Were health workers asked about FGM/C? If so, the results are not reported.

- “Unexplained pain” section should go under “Findings” section. Please, follow journal’s guidelines.

- There is no table or paragraph describing the demographic characteristics of participants. A paragraph in the methods section reports age and origin of women, I suggest this is moved to Findings. Also, characteristics of the health workers should be described.

- Lines 486-496: this paragraph does not report results, it discusses them. It would be better to move it to the discussion section.

- “Hysteresis and match” section should also be moved to the discussion.

Tables/supplementary materials:

- I suggest Tables 1 and 2 are included in the main body of the manuscript, and not as supplementary materials. The information they give is fundamental to understand the article.

- There are two supplementary materials called S1 (S1 Table, S1 appendix).

- Interview guides for health workers are not provided. Please include them as supplementary files as the guides for women.

General comments:

- Please, review English writing carefully. Improve orthography and style.

- All abbreviations and acronyms should be spelt out the first time they appear in the text (same applies to the abstract).

- Consider writing SSA women without hyphenation.

- Be consistent across the text with technical terms (e.g. bio medical or biomedical, use always the same).

6. PLOS authors have the option to publish the peer review history of their article (what does this mean?). If published, this will include your full peer review and any attached files.

Reviewer #1: No

---

## [Author Response · Author response to Decision Letter 0]

13 Jul 2021

1. Please ensure that your manuscript meets PLOS ONE's style requirements, including those for file naming. The PLOS ONE style templates can be found at https://journals.plos.org/plosone/s/file?id=wjVg/PLOSOne_formatting_sample_main_body.pdf andhttps://journals.plos.org/plosone/s/file?id=ba62/PLOSOne_formatting_sample_title_authors_affiliations.pdf <about:blank> 

Reply:Done. 

Headings and subheadings are changed so that they match the style requirements. The content wihin the subsections are following the formal demands. Table 1 and 2 are included in the text as suggested by reviewer 1. The interview guides are placed as S1 appendix and S2 appendix supporting information. File names follows the Plos One Style. A Material and Method section is established as suggested by reviewer #1, containing two subsections 1. Analytical frameworks 2. Study design and procedure

Quotation from informants are presented in the same style throughout the text.

The text has gone through proof reading, so the English language should now be much improved. Text are moved from background to the introduction, so that the introduction better is following the style of introductions. 

Respons to Comments written on the pdf file of the submitted reviewed manuscript.

Comment 1. Please consider a more informative title:

Reply:

The title has been changed: The former subtitle has been change into main title: “Health workers and Sub Saharan African women’s understanding of equal access to healthcare in Norway.”

Comment 2.The abstract must follow the journal style. This is too confusing and repetitive.

Reply: The abstract is changed to follow the journal style.

Comment 3. The intro is confusing, please follow the journal format and instructions

Reply: The introduction is changed. It starts with previous paragraph 4 as suggested with Reviewer #1, and follows the journal format: review of key literature and presentation of disagreements, aims and research questions. The text transferred from “Equal access to health care a theoretical background” to the Introduction starting on line 70 in the resubmitted version, has been shortened. 

Line 109 – 146 in older version that has been moved to Introduction now starts from line 70 - 105 in the new resubmitted version

In old submitted version: Line 111 is removed, as well as line 119 -122 has been removed, and lines 131 to 132. 

The aims are formulated in line 105 and research questions placed thereafter, line 108 -112

Comment 4: Which framework is the paper following? 

Reply: Candidacy theory and Bourdiues analytical frame are presented within a new Materials and methods section, subsection: Analytical frameworks. This is done by following suggestions from Reveiwer #1. The integration of the two theories are visible within the two tables, table 1 in line 305, and table 2 in line 390, following the suggestions from Reviewer #1.The two theories are also introduced in the Introduction line 113 -115

Comment 5. The intro and two subsequent sections could be shortened and presented more cohesive to make an argument for the study aims and rationale. Right now the study aim are getting lost

Reply: A couple of sentences that has less relevance to the discussion has been removed to shorten the article. It goes for sentence 111, 131 – 132, and sentence 137 to 138. The sentence 152 -153 has been removed, as well as 174 -175. These sentences are within the earlier version submitted. The sentences are invisible, of course, in the new version submitted. 

The aim of the study and guidelines for reading has been strengthened in the introduction and in the Material and methods section. 

Comment 6. The method must follow a format. Design, population, sample, data collection, measurments, statistical analyses etc.

Reply: This is not a quanitative study that require a statistical analyses. It is a qualitative study. The Reviewer # 1, has made suggestions, where information about purposive sampling, description of interviews and questions are asked for as well as the procedure of recruitment. The author has followed suggestions from reviewer #1. 

Reply: What I have written about financing in the text line 67 -70 is correct: “The article is part of a project called “Physical an psychological health care for girls and women with Female Genital Mutilation /Cutting that has been financed by the Norwegian Research Council and Norwegian Center for Violence and Traumatic Stress Studies with support from the Ministry of Justice”. 

A grant from the Norwegian Researh Council was given number: 262757. However it is the name of Norwegian Research Council that is the problem. In Norwegian it is Norges Forskningsråd. 

The title Norges Forskningsråd will be kept.

Grant number from the Ministry of Justice does not exist: 

The funding from NKVTS is not registered as a grant. 

The financial Disclosure will now read:

“Norges Forskningsråd has financed the main part of the research, grant number 262757.

 NKVTS with support from the Minstry of Justice has contributed to the financing of the research.

No – No funders had no role in study design, data collection and anlysis, decision to publish, or preparation of the manuscript”

Reply: The main funder of the research is Norges Forskningsråd, grant number 262757. However, Norwegian Centre for Violence and Traumatic Stress Studies has also contributed some financial support, and NKVTS again gets funding from the Ministry of Justice hat has no grant number. The money was allocated from chapter 291 post 21 within the State Budget.

 Reply: I have made a mistake here when it comes to data access and repository information. This is a qualitative data collection and it will not be able to share the data. Informants have not been informed as well, and it is not ethical to share data without their consent. In addition the data will not be available in a repository with a DOI.

I need to make changes in the Availability statement.

All relevant data are within the manuscript.

4. We note you have included a supporting information table to which you do not refere in the text of your manuscript. Please ensufe that you refer to Table S3 in your text, if accepted , production will need this reference to link the reader to the Table. 

Reply: There is no S3 Table. The sentence is removed. The two existing tables have been included in the text. There are now two interview guides called S1 Appendix.( This is interview guide for health workers), and S2 Appendix. (This is interview guide for SSA women).

5. Is the manuscript presented in an intelligible fashion and written in standard English?

Reply: The reviewer has answered no to this question. The text has been improved by professional proof readers.

Reviewer # 1

1. Abstract: I suggest the abstract follows a common structure (background, methods result and discussion) instead of including many subsections with specific titles). 

Reply: The abstract has been changed to follow guidelines and the subsections are removed.

2. Background: - Please, read journal’s instructions on the information that background section should include. The structure of the background (introduction) is not clear. 

a. I suggest starting with the 4th paragraph and ending the section by explaining the project in which the articles is included, and the objectives.

Reply: The section “Introduction” starts with the 4th paragraps.

Line 109 – 146 in older version has been moved to Introduction in new resubmitted version from line 70 - 105 .

The text has been shortened. In old submitted version: Line 111 is removed. 

Line 119 -122 has been removed, as well as lines 131 to 132.

The aims are more clearly formulated and research questions presented in the end of introduction.

b. -The introduction presents results, please remove them from this section.

Reply: Results have been removed. Line 91 -95 in old submitted version is removed

c. It is not necessary to present and reference other articles published from the same project in the background (introduction).

Reply: Articles and references from the same project is removed.

Methods

3. - Following journal’s instructions, the manuscript should contain sections on background, methods, results and discussion. The sections called “Equal access to healthcare” and “Habitus and hysteresis…”, thus, should be placed either in the background or in the methods section.

4. - My suggestion: create a general section called “materials and methods” where you can include these two sections on theory, then a subsection on “study design and procedures” (now called “Method”).

Reply: The suggestions by the reviewer #1 has been followed. The first theoretical background information is moved to the Introduction line 70 – 105. The theoretical framework (Candidacy and Bouridieus concepts) is placed within a new section called “Materials and Methods as suggested by #1 Reviewer. This Matherials and Methods section have two subsections 1) “Analytical frameworks” where the two theories used in the article are described, and 2) “Study design and procedure section”, where methods information about sample and recruitment is presented.

5. I do not think it is necessary to report the age of the social anthropologists.

Reply: The age is removed

6. - Inclusion criteria and procedures to approach health workers for interview are not explained. 

 Where were they recruited? How?

 Reply: A new paragraph has been added describing the procedure of recruitment.

 It starts line 227 -241, and 249 -259 

7. Paragraph starting in line 266 reports information that should be placed in the results section.

 Reply: The sentence is moved to the Result section in new version line 331

 Information about age in line 266 (old version) is moved to Result section 331-335

8. - I suggest to delete this sentence: “Since she had done fieldwork among Somalis and Gambians in 274 earlier projects [32,33,34,35]”. It is unnecessary to justify who performed the interviews, and to reference all their publications.

 Reply: The sentence is deleted

9. - Information on purposive sampling should be better placed at the beginning of the methods section for the sake of clarity (now in line 281).

Reply: Information about sampling is put in the beginning of the now “Study design and proce11dure section” from line 227

10. - It would be interesting to briefly describe the interview guides to help the reader understand which topics were covered in the interviews.

Reply: Description of the interview and questions asked is placed in line 224 -226. The interview guides are presented as supporting information in S1 Appendix and S2 Appendix

Findings:

11. - Tables 1 and 2 are cited in the text but they are supplementary files. As they are important to understand the results, I suggest including them in the main body of the article.

Reply: Both table 1 and 2 are included in the text. Table 1 from line 305. Table 2 from line 309.

12. - Line 338: a table is cited but it is not clear which one.

Reply: now this sentence is in line 319. The Number of the table is added table 1.

13. - Please, use the same style always when quoting.

Reply: Have used the same style everywhere when quoting and has used font size 10.

14 - Were health workers asked about FGM/C? If so, the results are not reported.

Reply: Material from the study is also published in other articles published in PLOS ONE. Here the focus is particularly on equal access to health care, and results is shown in the quotes from the women on their complain that they are not asked about FGM, and also in the section on unexplained pain. 

15. - “Unexplained pain” section should go under “Findings” section. Please, follow journal’s guidelines.

Reply: Unexplained pain is placed under the “Results “- section. The subtitle is changed from Finding to Result.

16. - There is no table or paragraph describing the demographic characteristics of participants. A paragraph in the methods section reports age and origin of women, I suggest this is moved to Findings. 

Reply: information about demographic charcertistics of SSA women has been moved to Result 331 -336 Information about demographic characteristics of health workers is placed in Results 264 -268

17. Also, characteristics of the health workers should be described.

Reply: Characteristics of health workers are described line 264 -268

18. - Lines 486-496: this paragraph does not report results, it discusses them. It would be better to move it to the discussion section.

Reply: This paragraph has been moved to the end of the discussion section, now line 514 -524 

19. “Hysteresis and match” section should also be moved to the discussion.

Reply: Hysteresis and match section has got a new Title: Discussion. The title Hysteresis and match is dropped. 

Tables/supplementary materials

20. - I suggest Tables 1 and 2 are included in the main body of the manuscript, and not as supplementary materials. The information they give is fundamental to understand the article

Reply: These tables have been moved to the text. Tables 1 in line 510, and Table 2 in line 611- 

21. There are two supplementary materials called S1 (S1 Table, S1 appendix). 

Reply: S1 table, is the table that now has been included in the text. S1 appendix, is the interview guide for health personel, which is placed as a supplementary material together with S2 appendix that is the interview guide for SSA women

22, - Interview guides for health workers are not provided. Please include them as supplementary files as the guides for women.

Reply: The interview guide for health workers are placed as a supplementary guide, as S1 appendix.

General comments:

23. - Please, review English writing carefully. Improve orthography and style.

Reply: The text has been sent to proof reading by professionals. 

24. - All abbreviations and acronyms should be spelt out the first time they appear in the text (same applies to the abstract).

- Consider writing SSA women without hyphenation.

Reply: Abbrevation and acronyms are spelt out first.

SSA women is written without hyphenantion

25. - Be consistent across the text with technical terms (e.g. bio medical or biomedical, use always the same).

Reply: biomedical is change to medical which is written consistently across text. Health care is written healthcare constantly in the text.

---

## [Editor Report · Decision Letter 1]

28 Jul 2021

Health workers and Sub Saharan African women's understanding of equal access to healthcare in Norway.

PONE-D-20-39751R1

Dear Dr. Lien,

We’re pleased to inform you that your manuscript has been judged scientifically suitable for publication and will be formally accepted for publication once it meets all outstanding technical requirements.

Kind regards,

Luisa N. Borrell, DDS, PhD

Academic Editor

PLOS ONE
---

## [Editor Report · Acceptance letter]

31 Aug 2021

PONE-D-20-39751R1 

Health workers and Sub Saharan African women’s understanding of equal access to healthcare in Norway

Dear Dr. Lien:

I'm pleased to inform you that your manuscript has been deemed suitable for publication in PLOS ONE. Congratulations! Your manuscript is now with our production department. 

Kind regards, 

on behalf of

Dr. Luisa N. Borrell 

Academic Editor

PLOS ONE